# Bioinspired polymeric supramolecular columns as efficient yet controllable artificial light-harvesting platform

Bin Mu [1], Xiangnan Hao[1], Xiao Luo[1], Zhongke Yang[1], Huanjun Lu[2] & Wei Tian [1] ✉

Light-harvesting is an indispensable process in photosynthesis, and researchers have been exploring various structural scaffolds to create artificial light-harvesting systems. However, achieving high donor/acceptor ratios for efficient energy transfer remains a challenge as excitons need to travel longer diffusion lengths within the donor matrix to reach the acceptor. Here, we report a polymeric supramolecular column-based light-harvesting platform inspired by the natural light-harvesting of purple photosynthetic bacteria to address this issue. The supramolecular column is designed as a discotic columnar liquid crystalline polymer and acts as the donor, with the acceptor intercalated within it. The modular columnar design enables an ultrahigh donor/acceptor ratio of 20000:1 and an antenna effect exceeding 100. Moreover, the spatial confinement within the supramolecular columns facilitates control over the energy transfer process, enabling dynamic full-color tunable emission for information encryption applications with spatiotemporal regulation security.

Efficient capture, transfer, and storage of solar energy represent a challenging area of research in the present day[1–3]. Inspired by natural process of photosynthesis, artificial systems have been developed to achieve effective harvesting of light energy through Förster resonance energy transfer (FRET) events from donor (absorbing) to acceptor (emitting) molecules[4–10]. These systems are commonly synthesized by anchoring donor/acceptor chromophores to various structural scaffolds, such as vesicles[11–13], micelles[14–20], nanoparticles[21–30], nanocrystals[31], gels[32–34], and biohybrid assemblies[35–39]. To ensure efficient light-harvesting capability, one primary objective is to increase the donor/acceptor ratio as much as possible so that more donors can transfer excitation energy to one acceptor. However, achieving this goal is complicated by the requirement for excitation energy to travel a longer diffusion length within the donor matrix to reach an acceptor. This is due to the distance-dependent nature of dipole-dipole dominated FRET, resulting in a decrease in efficiency with increasing distance in high donor/acceptor ratio systems[40–43]. Therefore, the

development of assembly platforms for light-harvesting that can provide efficient yet controllable energy transfer (ET) pathways is desirable.

It is well-established that the assembly mode of purple photosynthetic bacteria provides a natural prototype for exploring alternate pathways of ET. Specifically, bacteriochlorophyll (BChl) *a* stacks into curved columns along a ring-shaped axis with the aid of polypeptides and carotenoids (Fig. 1a)[44,45]. The interaction between BChl *a* within the column allows for the formation of delocalized electronic excitations, leading to increased exciton energy diffusion rates and distances that facilitate ET to the reaction center[46–48]. Drawing inspiration from this natural process, we seek to organize chromophores into densely packed arrays to mimic the excitation ET process. In this regard, supramolecular columns with efficient intracolumnar electronic interactions may be ideal candidates, although the molecular arrangement between the chromophores and resulting columnar morphology differs from that in natural systems. The modular

[1]Shanxi Key Laboratory of Macromolecular Science and Technology, Xi'an Key Laboratory of Hybrid Luminescent Materials and Photonic Device, MOE Key Laboratory of Material Physics and Chemistry under Extraordinary Conditions, School of Chemistry and Chemical Engineering, Northwestern Polytechnical University, Xi'an 710072, China. [2]Jiangsu Key Laboratory of Micro and Nano Heat Fluid Flow Technology and Energy Application, School of Physical Science and Technology, Suzhou University of Science and Technology, Suzhou 215009, China. ✉e-mail: happytw_3000@nwpu.edu.cn

Purple photosynthetic bacterial-based natural LHS

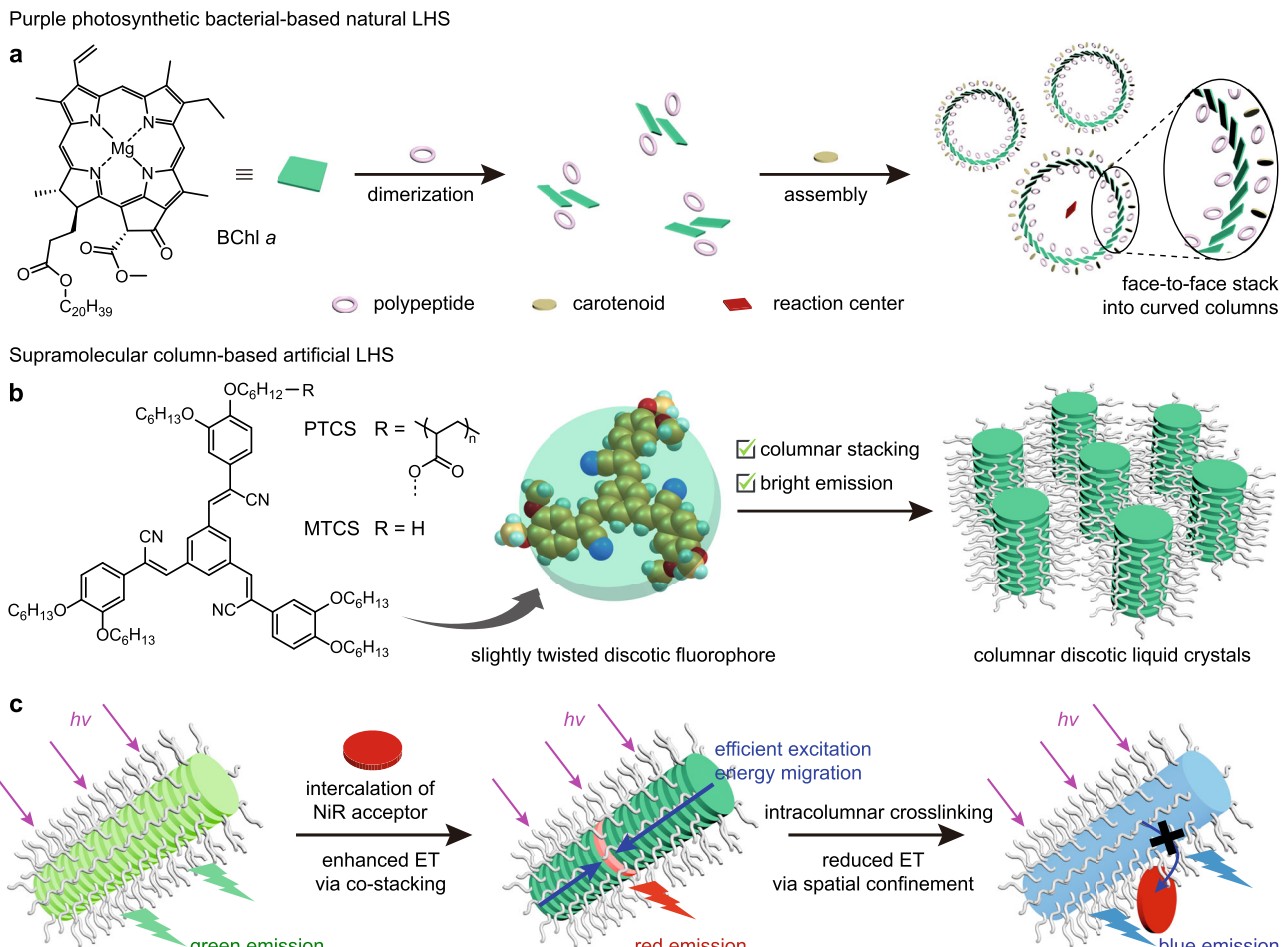

Supramolecular column-based artificial LHS

**Fig. 1 | Construction of supramolecular column-based LHS and its natural prototype. a** Schematic illustration depicting the supramolecular assembly of bacteriochlorophyll (BChl) *a* in purple photosynthetic bacteria. **b** Molecular structures of PTCS and MTCS, which are utilized to construct supramolecular columns. The slightly twisted discotic geometry of the TCS fluorophore facilitates self-assembly into columnar LC phases and preserves bright emission. **c** Modular columnar assembly enables efficient and controllable ET behaviors. The neat PTCS supramolecular columns exhibit bright green light emission. Co-stacked PTCS-NiR supramolecular columns display red emission with enhanced ET character due to the efficient excitation energy migration along the columns. Intracolumnar cross-linking of TCS fluorophores leads to blue fluorescence by reducing ET via spatial confinement.

columnar assembly approach offers flexibility in preparing multi-component architectures that facilitate ultrahigh donor/acceptor ratios in dynamic nanostructures. Importantly, the electronic coupling of stacked aromatic chromophores in these supramolecular columns generates exciton energy diffusion, enabling the rapid migration of excitation energy over a large distance along the columnar donor arrays before being transferred to the acceptor, where it is emitted. Although supramolecular columns hold significant potential for optimizing ET efficiency in mimicking the light-harvesting process, this effectuation has not yet been demonstrated yet.

In this study, we propose an approach to constructing an efficient light-harvesting system (LHS) by modular assembly of donor/acceptor chromophores in supramolecular columns. The ET efficiency in this system can be controlled by tuning the intracolumnar structures. To overcome excited-state quenching upon aggregation, we designed a slightly twisted non-planar discotic tricyanotristyrylbenzene (TCS) and introduced it as pendant groups to a polyacrylate backbone (Fig. 1b). The resulting side-chain discotic polymer (PTCS) exhibited columnar assembly in liquid crystal (LC) while maintaining green fluorescence with a high quantum yield ($\Phi_F$). Upon intercalation of the acceptor Nile red (NiR), the resulting modular supramolecular columns exhibited proper inter-chromophore electronic interactions, leading to a highly efficient LHS (Fig. 1c). Notably, the ET was detectable even at a high donor/acceptor ratio of 20,000:1, and the antenna effect (AE, a factor that describes how much brighter the acceptor emits when excited by the donor instead of being directly excited) was ultrahigh, exceeding 100. Furthermore, the supramolecular columnar assembly allowed for additional control over the ET process. The modularly co-stacked columns exhibited red emission with enhanced ET character, whereas intracolumnar crosslinking gradually reduced ET by expelling NiR from the columns, resulting in blue fluorescence (Fig. 1c). As a result, a dynamic full-color tunable red-green-blue (RGB) emission system was established, including pure white-light emission with $\Phi_F$ up to 0.28. This system demonstrated multi-level information encryption applications with spatiotemporal regulation ability.

## Results

### Construction of supramolecular column-based LHS

The PTCS was synthesized via an indirect polymer analogous method using trans-esterification of mono-hydroxyl-functionalized TCS (Supplementary Figs. 1–5) and poly(pentafluorophenyl acrylate)[49]. This method ensured almost quantitative efficiency (Supplementary Figs. 6–9), facilitating precise determination of the amount of TCS units incorporated into the PTCS. To illustrate the assembly behavior of PTCS and study the effect of polymerization on light-harvesting performance, a control compound, MTCS, with a symmetrical

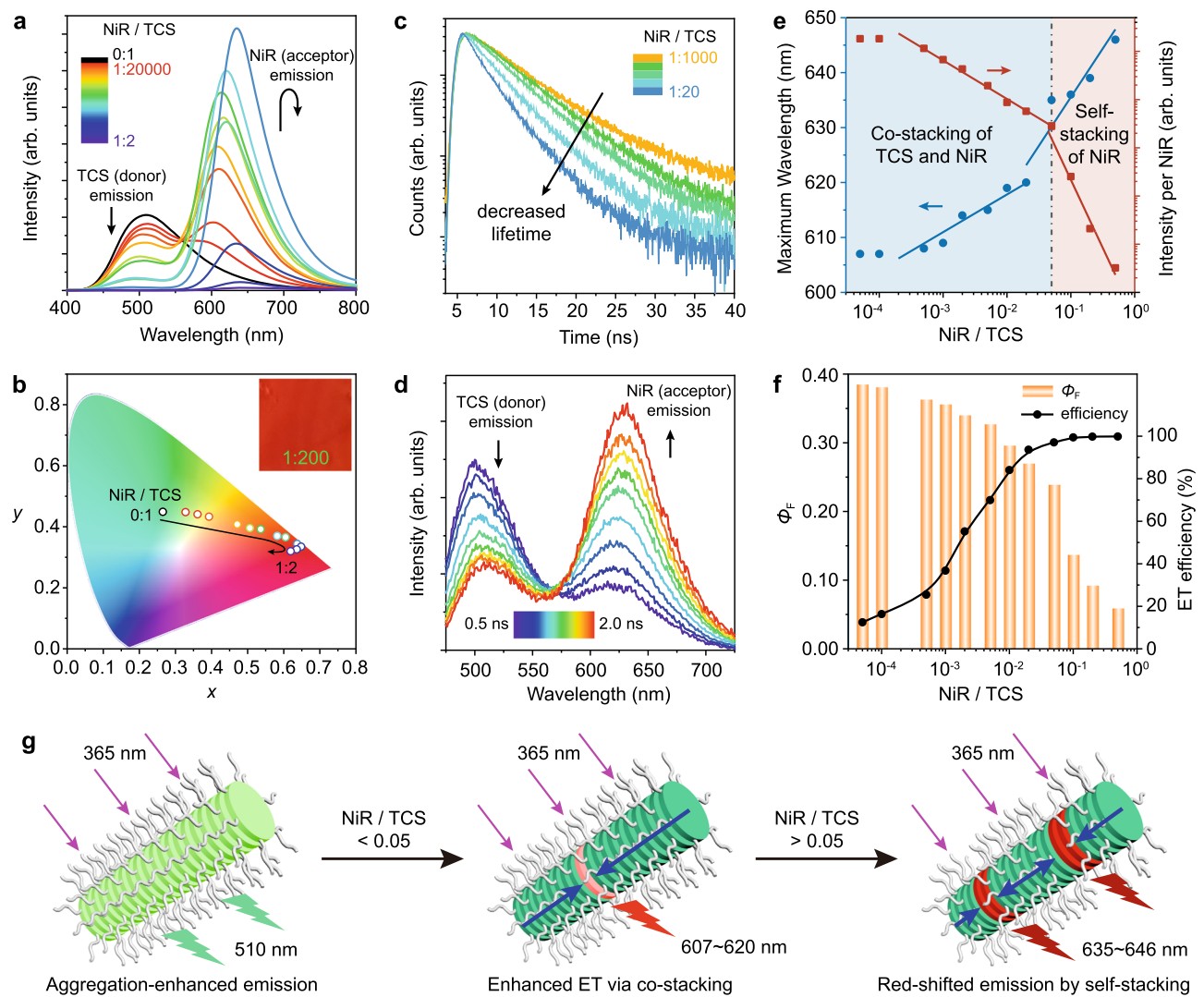

**Fig. 2 | High efficiency of supramolecular column-based LHS. a** Fluorescence emission spectra of PTCS-NiR with excitation at 365 nm. **b** The corresponding CIE 1931 chromaticity diagram illustrating the changing trend with increasing NiR/TCS ratio. **c** Fluorescence lifetime decay profile at the donor TCS emission of PTCS-NiR. **d** Time-resolved fluorescence spectra of PTCS-NiR after excitation at 365 nm, with a NiR/TCS ratio of 1:200. **e** Dependence of the maximum emission wavelength arising from the NiR and the relative intensity per NiR unit as a function of the NiR/TCS ratio. **f** Dependence of $\Phi_F$ and ET efficiency as a function of the NiR/TCS ratio. **g** Illustration of the modular assembly process for the supramolecular column-based LHS, where light is absorbed by TCS and transported with high efficiency to NiR, leading to NiR fluorescence emission. Neat PTCS supramolecular columns exhibit aggregation-enhanced emission characteristics. PTCS-NiR display enhanced ET character with a NiR/TCS ratio below 0.05, while self-stacking of NiR leads to red-shifted emission and reduced intensity when the ratio exceeds 0.05.

substitution was prepared (Fig. 1b). Both PTCS and MTCS formed thermotropic hexagonal columnar ($Col_h$) LC phases due to the discotic shape of the TCS core with aliphatic surroundings, which was reminiscent of the columnar structures generated by TCS derived LC compounds[50,51]. The temperature range of PTCS was significantly expanded across room temperature from below 0 °C to 123 °C, compared to MTCS that exhibited a LC range between 80 °C and 110 °C (Supplementary Fig. 10). The $Col_h$ phases were identified by their textures under polarized optical microscopy, with a pseudo focal conic fan-shaped texture for MTCS and an atypical texture for PTCS due to the small domains of LC polymers (Supplementary Fig. 11). The detailed packing structures were identified from the 1:√3 ratio of reciprocal small-angle X-ray spacings, characteristic of two-dimensional $Col_h$ lattices (Supplementary Figs. 12–14).

The PTCS displayed bright green fluorescence ($\Phi_F = 0.39$) in the columnar LC state, effectively enabling ET to the intercalated acceptor chromophores within the modular supramolecular columns. The solid-state emission of PTCS was preserved due to the aggregation

enhanced emission effect, a consequence of the slightly twisted geometry and the out-of-plane twisted configuration of the cyano groups, preventing tight π-overlapping of the inner TCS (Fig. 1b)[50–54]. Both absorption and emission spectra of PTCS experienced a bathochromic shift in the solid state compared to the solution state (Supplementary Fig. 15), indicating J-type aggregation of the discotic TCS units within the columns. Notably, the absorption spectrum of commercially available NiR overlapped well with the emission spectrum of PTCS (Supplementary Fig. 16), facilitating efficient ET from PTCS to NiR when co-stacked within the supramolecular columns. As demonstrated in Fig. 2a, an increase in the NiR/TCS ratio led to a decrease in TCS emission intensity at 510 nm and a corresponding rise in NiR emission around ~620 nm when excited at 365 nm (fluorescence spectra at excitation 530 nm are presented in Supplementary Fig. 17). The CIE 1931 chromaticity diagram confirmed the track of the emission color switching followed by the variation of NiR/TCS ratios (Fig. 2b). Time-resolved fluorescence experiments further validated the ET process, evidenced by the decrease in the TCS emission lifetime at

510 nm with increasing NiR content, indicating rapid ET process (Fig. 2c). At a selected ratio of 1:200, time-resolved fluorescence spectra revealed a decrease in donor emission at 510 nm within 2 ns, accompanied by an increase in acceptor emission at ~620 nm (Fig. 2d). These results demonstrate efficient excitation energy transfer from TCS to NiR within the co-stacked supramolecular columns. Additionally, fluorescence spectra recorded under varying excitation power revealed a linear intensity dependence for both donor and acceptor emissions with a slope of about 1 throughout the entire excitation range (Supplementary Fig. 18). This linear dependence proves that ET in the supramolecular column-based LHS occurs due to a single photon event, eliminating the possibility of multiple exciton processes. Notably, the ET was clearly detectable even at a trace amount of NiR with a high donor/acceptor ratio of 20,000:1. This observation underscores the efficient transfer of light energy absorbed by the donor TCS units over substantial distances within the supramolecular columns, ultimately reaching the acceptor NiR, where it is emitted.

The modular columnar assemblies of PTCS-NiR were corroborated by intensive study of their structural and photophysical properties at different NiR/TCS ratios. X-ray scattering analysis revealed the retention of $Col_h$ at lower NiR/TCS ratios. However, at higher ratios, the intercolumnar hexagonal symmetry was disrupted due to the swelling of the co-stacked columns (Supplementary Fig. 19). To assess the performance of this system as an artificial LHS, the maximum emission wavelength from NiR and the relative intensity per NiR were plotted against the NiR/TCS ratio, revealing two distinct stages (Fig. 2e and Supplementary Fig. 20). In the first stage (NiR/TCS < 0.05), NiR molecules were individually intercalated into PTCS columns, resulting in a gradual red-shift of the emission wavelength and a slight decrease in intensity with increasing NiR amount. The marginal alterations observed in the spectra were attributed to weak electronic interactions from the co-stacking of NiR and TCS in columns. In the second stage (NiR/TCS > 0.05), self-stacking of NiR occurred, leading to a prominent red-shift in wavelength and an abrupt decrease in intensity. This process also correlated with an increase in quantum yield of ET (or ET efficiency), reaching nearly 100%, while the $\Phi_F$ gradually decreased from 0.39 to 0.06. Both trends exhibited a comparable inflection point around a NiR/TCS ratio of 0.05 (Fig. 2f). These findings, depicted in Fig. 2g, clearly elucidate the assembly mode of the modular supramolecular columns and their emission characteristics, which is challenging to achieve in conventional LHSs due to the absence of well-defined assembly structures[14–38]. Overall, these results not only unravel the mechanism underlying the evolution concerning varying NiR/TCS ratios but also contribute to the modular assembly mode in supramolecular columns.

## Efficient excitation ET within the supramolecular columns

Next, we seek to gain an in-depth insight into the supramolecular columnar assemblies that constitute the primary contribution to the efficient light-harvesting efficiency. Supramolecular columnar assemblies of these discotic chromophores are expected to greatly decrease the inter-chromophore distances and thus favor efficient excitation ET, which can be evidenced by a fluorescence anisotropy decrease[27,55]. To verify this conjecture, fluorescence anisotropy experiments were conducted on PTCS in various states (Fig. 3a). When PTCS was dispersed in polymethyl methacrylate (PMMA), the columnar assembly was disrupted, leading to restricted molecular motion and an anticipated high anisotropy. Conversely, a well-ordered columnar assembly with appropriate intracolumnar electronic coupling was expected to facilitate efficient excitation ET, resulting in substantially reduced anisotropy. Indeed, experimental results demonstrated a 15-fold reduction in fluorescence anisotropy from 0.24 to 0.016 within the columnar assembly (Fig. 3b and Supplementary Figs. 21–22). The intercalation of NiR into these supramolecular columns accelerated the migration of excitation energy within the system, leading to a

remarkable decrease in anisotropy to an impressive 0.006. However, it is worth noting that achieving such a low anisotropy level poses a challenge in precisely defining its further decrease with increasing NiR/TCS ratios due to instrumental limitations. The nearly zero fluorescence anisotropy exhibited a larger standard error than the observed value, suggesting that the excitation energy redistributed rapidly within the columns. Moreover, the exciton migration rate constant was estimated to be $1.65 \times 10^{11}$ L mol$^{-1}$ s$^{-1}$ (Supplementary Fig. 23), indicating significantly faster migration of exciton energy within the columns compared to the diffusion limit for bimolecular reactions in solution[31,56]. The supramolecular columnar assembly mode was further elucidated by density functional theory (DFT) calculations (Fig. 3c). The optimized MTCS trimer demonstrated weak π-stacking (π-π distance: 3.4 Å) of the non-planar discotic TCS cores and nanosegregation with bulky alkyl substitutions, favoring intracolumnar delocalization of electronic excitation and fast exciton energy migration along the columns[46–48]. Moreover, NiR was efficiently intercalated into the MTCS columns, with an average π-stacking spacing of 3.4 Å, enabling effective ET from MTCS to NiR. Therefore, the modular columnar structures demonstrated remarkable capabilities in facilitating fast exciton energy migration among donor arrays and ensuring efficient ET to acceptors, thus enabling additional donors away from the acceptor to feed excitation energy to the acceptor through columnar pathways.

Inducing liquid crystallinity in the supramolecular columns improves light-harvesting efficiency, as demonstrated by a comparative analysis of the light-harvesting performance of MTCS in LC and crystalline states. MTCS exhibited columnar structures in both LC and crystalline states: a $Col_h$ LC phase from 80 to 110 °C and a rectangular columnar crystalline phase below 80 °C (Supplementary Fig. 12). Both columnar phases were suitable for constructing supramolecular column-based LHSs. The light-harvesting studies revealed that MTCS-NiR exhibited higher AE values and enhanced ET efficiency in the LC state compared to the crystalline state, particularly at low NiR/TCS ratios (Fig. 3d and Supplementary Figs. 24–27). Specifically, the AE for crystalline MTCS exhibited an initial increase and subsequent decrease as the NiR/TCS ratio decreased, while liquid-crystalline MTCS showed a consistent, monotonic increase up to approximately 60. Additionally, the efficiency was further measured by the average number of donor molecules quenched by a single acceptor ($K_{sv}$), which was obtained through linear curve-fitting of the Stern-Volmer plot of the donor as a function of the NiR/TCS ratio (Fig. 3e and Supplementary Fig. 28)[57–59]. The $K_{sv}$ value for MTCS in the LC state (425) significantly surpassed that observed in the crystalline state (282). Notably, temperature-dependent experiments excluded the possibility of temperature effects on light-harvesting efficiency (Supplementary Figs. 29–31). Thus, the light-harvesting performance was enhanced via the induction of liquid crystallinity, which may be explained by the dynamic ordering of LCs. The ordered packing structures were found to be crucial for efficient excitation ET, and any imperfection in the crystal could disrupt energy diffusion, whereas soft LCs with dynamic defect self-repair behavior offer a promising solution to avoid defect-induced interruptions in energy transmission[60–62].

Moreover, the incorporation of polymer main chain induces intercolumnar correlation within the supramolecular columns, a significant factor contributing to the light-harvesting efficiency. This can be demonstrated by a comparative study of PTCS and MTCS in the LC state. In particular, PTCS exhibited an AE value exceeding 100 when the NiR/TCS ratio was smaller than 1:2000 (Fig. 3d), along with a $K_{sv}$ value of 675 (Fig. 3e), both of which clearly surpassed MTCS. The presence of polymer main chain-induced columnar superlattice assembly appears to be responsible for the observed enhancement in light-harvesting efficiency. PTCS and MTCS exhibit two distinct types of $Col_h$ phases, characterized by different hexagonal lattice parameters, despite sharing the same discotic mesogens (Fig. 3f). The lattice parameter of MTCS was determined to be $a = 2.68$ nm, closely

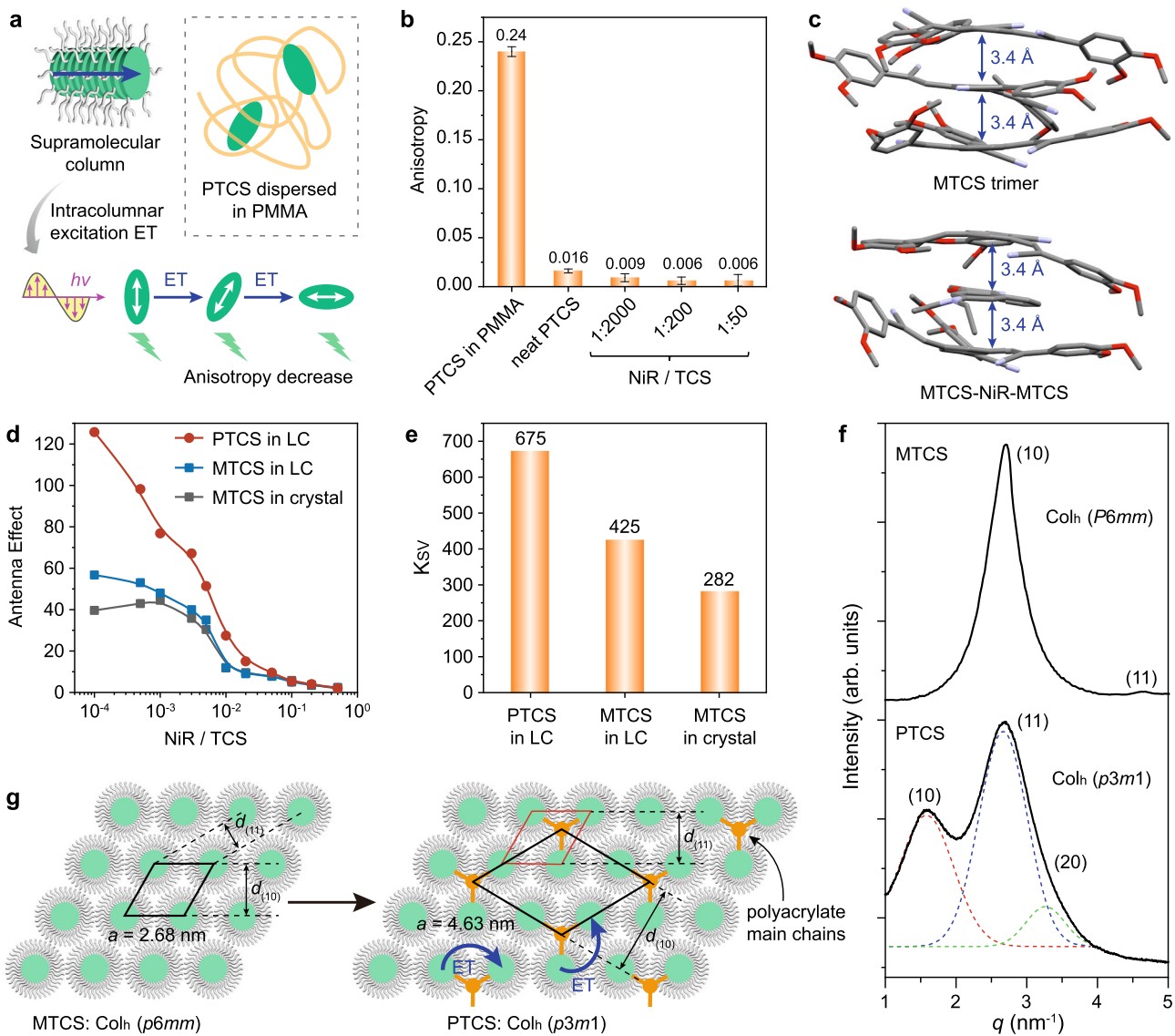

**Fig. 3 | Mechanism underlying the efficient ET in the supramolecular columns.**
**a** Schematic illustration depicting the fluorescence anisotropy decrease due to efficient excitation ET along the supramolecular column. **b** Steady-state anisotropy for PTCS in PMMA, neat PTCS and PTCS-NiR with varying NiR/TCS ratios. Error bars represent standard error of the mean. **c** Optimized geometries of MTCS trimer and MTCS-NiR-MTCS species, where the peripheral long alkyl chains are omitted for clarity. **d** Antenna effect of PTCS-NiR and MTCS-NiR in both LC or crystalline states as a function of NiR/TCS ratio. **e** $K_{sv}$ of PTCS-NiR and MTCS-NiR in LC or crystalline states. **f** X-ray scattering profiles of MTCS and PTCS in the LC state, with proposed indexing based on $Col_h$ ($p6mm$) and $Col_h$ ($p3m1$) phases, respectively. Dashed lines indicate resolved peak components. **g** Models of the two-dimensional columnar arrangement of MTCS and PTCS in $Col_h$ ($p6mm$) and $Col_h$ ($p3m1$) phases, respectively. The black parallelograms represent the crystallographic unit cells.

matching the molecular dimension (Supplementary Fig. 13), thereby supporting the proposition of a $Col_h$ structure with $p6mm$ symmetry, similar to related columnar discotic systems (Fig. 3g)[50,51]. Additionally, the electron density map confirmed the formation of columns via a helical stacking manner of three-armed MTCS molecules (Supplementary Fig. 14). In contrast, the lattice parameter of PTCS ($a = 4.63$ nm) was found to be approximately √3 times larger than that of MTCS, and the (11) peak was much stronger than (10) peak (Fig. 3f). This indicated the formation of a hexagonal columnar superlattice. The number of molecules per unit cell was calculated to be three, implying that a three-column bundle may serve as the packing unit for columnar superlattice assembly. The columnar bundle was generated by the aggregation of main chains and further distribution on the basis of a hexagonal lattice (Fig. 3g). This packing arrangement reduced the symmetry to trigonal, resulting in a $p3m1$ plane group[63–66]. Therefore, the enhanced intercolumnar correlation resulting from covalently

linked columns should also account for the improvement in light-harvesting, as this correlation may facilitate intercolumnar ET (Fig. 3g), bearing more resemblance to natural photosynthetic antennas[44–48].

Overall, the obvious ET though in an exceptionally high donor/acceptor ratio of 20,000:1, together with an ultrahigh AE exceeding 100 and a $K_{sv}$ value of 675 for PTCS, outperforms the majority of recently reported artificial LHSs[11–39]. This performance is attributed to the polymer-assisted supramolecular columnar superlattice assembly in dynamic LC state. In particular, the modular supramolecular columnar assembly constitutes the primary contribution to the efficient light-harvesting efficiency by providing robust excitation ET pathways, enabling long distance exciton energy diffusion. This is in contrast to conventional LHSs based on classical FRET mechanism, which suffer from the critical limitation of a short distance between donor and acceptor chromophores. Additionally, the presence of liquid crystallinity and intercolumnar correlation has been shown to

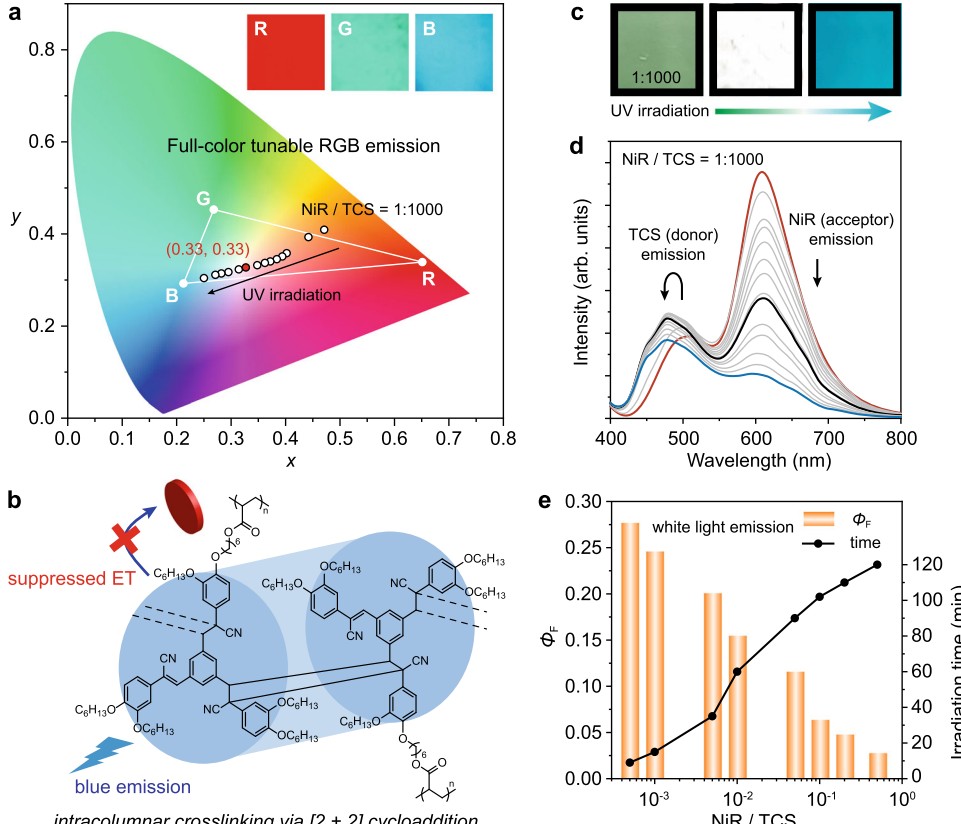

**Fig. 4 | Dynamic control of ET efficiency and full-color tunable emission. a** CIE 1931 chromaticity diagram illustrates the full-color tunable RGB emission, with insets showing the fluorescence images of typical RGB emission. The representative color switching trace of PTCS-NiR with a NiR/TCS ratio of 1:1000 under UV irradiation is also presented. **b** The proposed photochemical reaction under UV irradiation for the PTCS-based supramolecular column, leading to the suppressed ET process. **c** The corresponding color change of fluorescence images and white light emission. **d** Fluorescence emission spectra evolution of PTCS-NiR under UV irradiation (NiR/TCS = 1:1000). The black line corresponds to the white color coordinate (0.33, 0.33) represented by the red solid circle in (**a**). **e** The required irradiation time to achieve white light emission of PTCS-NiR and the corresponding $\Phi_F$ as a function of NiR/TCS ratio.

further promote the light-harvesting efficiency by optimizing the ET pathways.

## Control of ET process and concomitant multi-color emission

In addition to the benefit of high efficiency, the supramolecular columnar assembly offers an additional advantage of enabling control over the ET process through stimuli-induced spatial confinement, allowing for dynamic tuning of the emission in a full-color range (Fig. 4a). Actually, stimuli-responsive LHSs that yield dynamic color variations in a single nanostructured scaffold have been explored very little to date despite the tremendous potential for applications[4–39]. Given that cyanostyrenes are typical photo-responsive units[67–70], we seek to modulate the ET process of PTCS-NiR via light irradiation. PTCS underwent *Z-E* isomerization in solution state (Supplementary Figs. 32–33), while ordered packing assembly in the condensed state facilitated [2 + 2] cycloaddition (Supplementary Figs. 34–35). It is noteworthy that due to the columnarly stacked TCS surrounded by alkyl shells, light-induced cycloaddition was confined within the intracolumnar cores (Supplementary Fig. 36). Intracolumnar cross-linking led to the expulsion of NiR from the TCS columns, substantially reducing the ET process due to the absence of co-stacked structure (Fig. 4b). Consequently, despite the emission color for PTCS-NiR varying from green (G) to red (R) depending on the NiR/TCS ratios, complete crosslinking of PTCS was expected to yield blue (B) emission upon light irradiation (Supplementary Figs. 37–38). Selective cross-linking of PTCS-NiR (partial crosslinking and partial ET reactions) generated various emission colors within the RGB color system. As a

typical example of PTCS-NiR with a NiR/TCS ratio of 1:1000, fluorescence color gradually switched from green-yellow to white and eventually to blue upon ultraviolet (UV) irradiation (Fig. 4c). Fluorescence spectroscopy confirmed the reduction in NiR emission intensity around ~610 nm due to decreased ET from the disrupted co-stacked columnar structure (Fig. 4d). The emission wavelength of PTCS showed a blue-shift, progressing from 510 to 478 nm. The CIE 1931 chromaticity diagram confirmed the track of the emission color switching followed by UV irradiation, highlighting the cooperative control of cycloaddition and ET processes (Fig. 4a). Interestingly, pure white-light emission of (0.33, 0.33) in CIE coordinate was obtained with appropriate irradiation time, which was applicable across nearly the entire range of NiR/TCS ratios (Supplementary Fig. 39). However, the irradiation duration and emission efficiency varied with NiR/TCS ratios (Fig. 4e). Specifically, higher ratios required longer irradiation time, possibly due to increased hindrance from intercalated NiR, slowing down the cycloaddition process among TCS columns. The $\Phi_F$ of the white light emission decreased from 0.28 to 0.03 as the NiR/TCS ratio increased, largely due to self-quenching of NiR. Overall, light-triggered control of ET process in the supramolecular column-based LHS resulted in time-dependent color changes, promising programmable emission color tuning possibilities.

## Information encryption with spatiotemporal regulation ability

The dynamic multi-color tuning capability of this supramolecular column-based LHS makes it particularly suitable for multi-level information encryption applications with spatiotemporal regulation

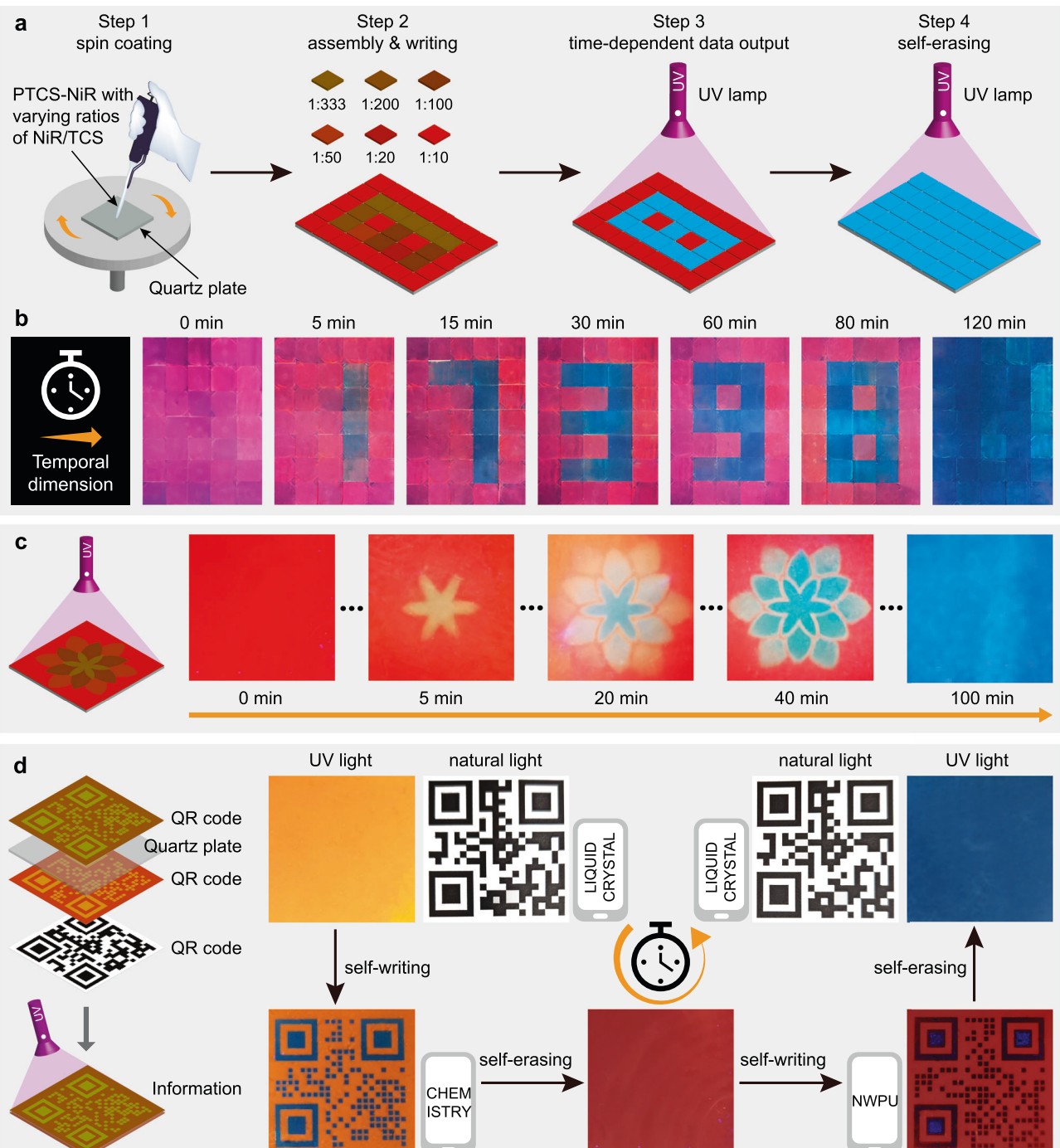

**Fig. 5 | Demonstration of spatiotemporally regulated information encryption.**
**a** Schematic illustration for fabricating fluorescent patterns via spin coating and
assembly, followed by data output and self-erasing under UV irradiation. **b** The
evolution of fluorescent images of the pattern under UV irradiation duration,
demonstrating time-dependent output of multiple pieces of information and self-

erasing. Specific information can be identified at designated intervals.
**c** Presentation of a flower-like fluorescent pattern gradually blooming and fading
under UV irradiation. **d** Decomposition of the multi-level features of the pattern
using triple QR codes, with information scanning from the QR code evolving over
time during UV irradiation.

security. This unique feature enables a single entity to store multiple
pieces of information, with the true piece of information only being
revealed at a specified time. To verify this hypothesis, we conducted
several experiments to demonstrate the practical applications of this
system. Firstly, a series of PTCS-NiR with varying NiR/TCS ratios were
fabricated as films on small quartz plates, which were then regularly
assembled into a big pattern composed of 35 independent films
(Fig. 5a). Under natural light, the information encoded in this pattern

was invisible. However, under continuous UV irradiation, the data
selectively emerged over time, displaying characters such as "1", "7",
"3", "9", "8", and eventually self-erasing (Fig. 5b). Secondly, a flower-like
fluorescent pattern was fabricated using a similar method, showcasing
a colorful flower that gradually bloomed and then faded away over
time (Fig. 5c). Thirdly, an information pattern with different quick
response (QR) codes on each side of a quartz plate was created.
Beneath it was another QR code panel (Fig. 5d). When exposed to

natural light, the QR code "LIQUID CRYSTAL" was easily read by a phone. However, under UV light, no information was observed. Through continuous UV irradiation, new information successively decoded, revealing the QR codes "CHEMISTRY" and subsequently "NWPU" before self-erasing. These successful demonstrations of multi-level information encryption with spatiotemporal regulation have significant implications for designing and fabricating high-level security information encryption systems, showcasing the versatility and potential of the supramolecular column-based LHS in practical applications.

## Discussion

In summary, we have developed a polymeric supramolecular column-based artificial LHS inspired by the natural light-harvesting of purple photosynthetic bacteria. This system is comprised of a columnar discotic liquid crystalline polymer PTCS, enabling efficient transfer of absorbed light energy from the donor PTCS to the acceptor NiR through a modular assembly of these chromophores in supramolecular columns. Notably, the achieved donor/acceptor ratio of up to 20,000:1 and an AE exceeding 100 highlight the exceptional light-harvesting efficiency. This remarkable efficiency is primarily attributed to the effective excitation ET within the supramolecular columns, which mimics the ring-shaped columnar assembly mode of BChl $a$ in purple photosynthetic bacteria. Furthermore, the presence of columnar liquid crystallinity and polymerization-induced intercolumnar correlation has been shown to enhance light-harvesting efficiency by optimizing the ET pathways. Additionally, the supramolecular columnar assembly enables control over the ET process via spatial confinement effect. This control is achieved by light-induced intracolumnar crosslinking, which expels the NiR from the TCS columns. The efficient yet controllable ET feature endows the system with dynamic full-color tuning capabilities on a time scale, demonstrating multi-level information encryption applications capable of spatiotemporal regulation. Overall, the modular assembly of donor/acceptor chromophores in supramolecular columns not only provides an avenue toward artificial photosynthesis applications but also offer a versatile methodology for constructing dynamic multi-color emissive materials.

## Methods

### Materials

Methyl 3,4-dihydroxybenzoate (98%, Innochem), 1-bromohexane (>98%, TCI), lithium aluminium hydride (>95%, TCI), trimethylsilyl cyanide (97%, Innochem), lithium tri-tert-butoxyaluminum hydride (97%, Aladdin), tetrabutylammonium hydroxide (40% in methanol, Innochem), 1,3,5-benzenetricarbonyl trichloride (99%, Innochem), benzene-1,3,5-tricarbaldehyde (>98%, TCI) and pentafluorophenyl acrylate (98%, TCI) were used as received.

### Measurements

The NMR spectra were recorded using Bruker Avance 400 instruments. For mass spectrometry (MS), we employed matrix-assisted laser desorption/ionization time-of-flight (MALDI-TOF) analysis, conducted on a Bruker Autoflex II instrument with 2,5-dihydroxybenzoic acid (DHB) as the matrix. Fourier transform infrared spectra (FTIR) were obtained using a Nicolet iS50 infrared spectrometer. Gel-permeation chromatography (GPC) measurements were performed at 25 °C using a Waters 515 system with a Wyatt Technology Optilab rEX differential refractive index detector and tetrahydrofuran as the eluent, at a flow rate of 1.0 mL min⁻¹. The molecular weights were calculated using a calibration curve based on polystyrene standards. Differential scanning calorimetry (DSC) thermograms were obtained using a Netzsch DSC 214 system under a nitrogen atmosphere at a flow rate of 20 mL min⁻¹. Polarized optical microscopy (POM) was conducted with a Nikon E400POL microscope, equipped with an Instec

HCS302 hot and cold stage. For X-ray scattering experiments, we used the beamline BL16B1 at the Shanghai Synchrotron Radiation Facility (SSRF) and an Anton Paar SAXSpoint 2.0 instrument, defining the scattering vector $q$ as $4\pi\sin\theta/\lambda$ with $\lambda$ being 0.1542 nm (Cu-$K_\alpha$ radiation) and $2\theta$ the scattering angle. Ultraviolet visible (UV-vis) absorption spectra were recorded on a Shimadzu UV-2550 spectrometer. For fluorescence analyses, including absolute quantum yield and time-resolved fluorescence lifetime, we used an Edinburgh FLS1000 spectrometer, employing a 365 nm picosecond pulsed excitation source for time-resolved measurements.

### Theoretical calculations

The geometries of all structures were optimized using density functional theory (DFT) calculations with Gaussian 09 program[71], utilizing the hybrid B3LYP functional and the 6-31G* basis set. Grimme's D3BJ dispersion correction was used to improve calculation accuracy[72].

### Synthesis

Full synthetic details and molecular characterization data are available in the Supplementary Information.

## Data availability

The data that support the findings of this study are available within the paper and its Supplementary Information, and also from the authors upon request.

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

## Acknowledgements

This work was supported by the National Natural Science Foundation of China (22175142, B.M.; 22022107, 22071197, W.T.), Shaanxi Fundamental Science Research Project for Chemistry & Biology (22JHQ019, B.M.), and the Fundamental Research Funds for the Central Universities (G2023KY0603, B.M.; D5000230114, W.T.). We thank the Analytical & Testing Center of Northwestern Polytechnical University for DSC mea-surement. We would also like to thank Beamline BL16B1 at Shanghai Synchrotron Radiation Facility (SSRF) for providing the beamtime.

## Author contributions

B.M. and W.T. conceived, designed, and directed the project. X.H., X.L., and Z.Y. performed the experiments. B.M., X.H., and H.L. analyzed the data. B.M. and W.T. wrote the paper. All authors discussed the results and commented on the manuscript.

## Competing interests

The authors declare no competing interests.
