## [Peer Review File · Nature Communications]

REVIEWER COMMENTS

Reviewer #1 (Remarks to the Author):

This work by Mu et al. introduces a new range of functional supramolecular materials exploiting a FRET process based on LC structures where a dye can intercalate to achieve red, blue or green colours, depending on the interaction of the dye with the LC assemblies.

The synthesis of the materials is well described and seems solid, with good evidence that the molecules of interest are indeed obtained. There is no doubt the luminescent study is very thorough and very convincing, showing the excellent luminescent properties of the systems. My main concerns with this publications are:

1/ I feel the authors should justify their choice of system better - the parallel with purple photosynthetic bacterial is to me very obscure, and the link seems to me very weak. Surely the authors can find examples of other, natural or synthetic, FRET systems based on supramolecular assemblies which exploit light harvesting to emit a range of colours. For instance, see:

Ahn, T. K.; Avenson, T. J.; Ballottari, M.; Cheng, Y.-C.; Ni-yogi, K. K.; Bassi, R.; Fleming, G. R., Architecture of a charge-transfer state regulating light harvesting in a plant antenna protein. *Science* 2008, 320 (5877), 794.

Wei, X.; Su, X.; Cao, P.; Liu, X.; Chang, W.; Li, M.; Zhang, X.; Liu, Z., Structure of spinach photosystem II–LHCII super-complex at 3.2 Å resolution. *Nature* 2016, 534 (7605), 69-74.

Son, M.; Pinnola, A.; Bassi, R.; Schlau-Cohen, G. S., The electronic structure of lutein 2 is optimized for light harvest-ing in plants. *Chem* 2019, 5 (3), 575-584.

Zhang, D.; Liu, Y.; Fan, Y.; Yu, C.; Zheng, Y.; Jin, H.; Fu, L.; Zhou, Y.; Yan, D., Hierarchical self-assembly of a dandelion-like supramolecular polymer into nanotubes for use as highly effi-cient aqueous light-harvesting systems. *Adv. Func. Mater.* 2016, 26 (42), 7652-7661.

Sarkar, A.; Behera, T.; Sasmal, R.; Capelli, R.; Empereur-mot, C.; Mahato, J.; Agasti, S. S.; Pavan, G. M.; Chowdhury, A.; George, S. J., Cooperative supramolecular block copolymeriza-tion for the synthesis of functional axial organic heterostruc-tures. *J. Am. Chem. Soc.* 2020, 142 (26), 11528-11539.

Song, Q.; Goia, S.; Yang, J.; Hall, J.C.L.; Staniforth, M.; Stavros, V.G.; Perrier, S. Efficient Artificial Light-Harvesting System Based on Supramolecular Peptide Nanotubes in Water, *J. Am. Chem. Sci.*, 2021, 143 (1), 382-389.

Kim, H.-J.; Nandajan, P. C.; Gierschner, J.; Park, S. Y., Light-harvesting fluorescent supramolecular block copoly-mers based on cyanostilbene derivatives and cucurbit[8]urils in aqueous solution. *Adv. Func. Mater.* 2018, 28 (4), 1705141.

2/ I am not too clear why comparing a single TCS structural molecule with a polymer-conjugated TCS, and why choosing a fluorinated acrylate? Stability of the structures? Solubility? Polarity?

3/ The characterisation of the assembled structures could be more thorough - although the authors demonstrate the LC properties, I am not too clear why they expect a stacked / columnar structure. Do they obtain individual rod-like nanoparticles in solution? Could the authors image these structures using TEM or AFM, or characterise them using scattering techniques such as SANS?

Overall I think the work is novel and could be published in Nature Comm, the luminescent properties are remarkable, but the authors should better justify their choice of materials, and better characterise the supramolecular structure they obtain.

Reviewer #2 (Remarks to the Author):

The authors present their studies in discotic molecular assemblies in facilitating long distance energy transfer inspired by light harvesting antenna in natural photosynthesis. They claimed that observation of large antenna effect in the assemblies and carried out variation of the acceptor/chromophore ratios as well as environmental modifications on the rigidity of the assemblies. They used time resolve fluorescence lifetime and anisotropy measurements to support their claims. Finally, they showed the chemically tunability of the emissions and proposed applications of these properties.

I recommend its publication of revised manuscript in a materials focused journal rather than Nature Communications for the reasons below:

1. The manuscript is not written with a high standard of scholarship with poor flow and repeated statements. Some of the concepts, such as antenna effect, were not defined. Some of the claims are based on hotchpotch results without firm and convincing analyses.
2. The application of discotic liquid crystal materials for light harvesting has been in the literature for decades. Thus, the concept is not novel. Peoples have used porphyrins, phthalocyanines, cronenes and other extended planar aromatic groups to form the columnar structure to conduct charges and facilitate energy transfer.
3. The authors need to have an excitation fluence dependent study to determine the “antenna effects” and energy donor/acceptor ratio to make sure that the effect is not due to multiple exciton generation in the system. It is important to quantify the quantum yield of energy transfer.
4. The authors were inspired by natural photosynthetic light harvesting systems with chlorophyll arrays, how ever they ignored an important aspect of this array where the chlorophylls are arranged with a

propeller fashion instead of well-aligned face-to-face conformation as in the columnar assemblies in their study. Many self-assembled systems are very sensitive to the alignment of the atoms between the chromophores. It has been well documented that for energy or charge transfer, some off-set between the pi-stacked entities of the same species could have superior properties in energy and charge transfer and transport. Although the authors indicated that they tried to avoid the quenching of the exciton by adapting a twisted conformation, but inter-chromophore interactions were not carefully characterized.

5. The authors claimed that “the electronic coupling of stacked aromatic chromophores in resulting supramolecular columns generates coherent energy diffusion” but the concept of coherent energy transfer was not elaborated and proven for the systems in the study.

6. The emission anisotropy measurements have serious flaws. In figure 3b, the authors compared the anisotropy of the isolated chromophore in a rigid PMMA matrix which has a much higher anisotropy due to a fixed orientation of the transition dipole moment for the emission to the chromophore with Nile red in columnar structure. It will be useful to just measure the columnar chromophore only without Nile red as a reference. According to the authors that the columnar structure has a spiral orientation between pi-stacked molecules. Thus, even without Nile red, it could have energy diffusion from one chromophore to another and have lower anisotropy compared to the isolated chromophore in the solid matrix. If the insertion of Nile red speeds up the energy migration following the Förster mechanism, the decay of anisotropy would be faster. Without the comparison with the neat columnar system, the study is in question. Meanwhile, I am not convinced on the small change in the system with Nile red. What are the error bars in those measurements?

Reviewer #3 (Remarks to the Author):

The authors present the construction of a supramolecular assemblies for mimicking natural light-harvesting system of purple photosynthetic bacterial. A discotic columnar liquid crystalline polymer was selected as an energy donor and NiR served as an energy acceptor. NiR intercalated into the polymeric columnar liquid crystal to form the supramolecular artificial light-harvesting system. The highly ordered packing of chromophores in such light-harvesting system based on liquid crystalline polymer facilitates the efficient energy transfer from energy donors to acceptors. It is impressive that the energy transfer can be observed even at ultrahigh D/A ratio of 20000:1. The emission colors of the light-harvesting systems can be tuned by changing the ratio of D to A. The light-harvesting systems with different emission color have been applied for information encryption with spatiotemporal regulation security. Overall, the results presented in this paper is of great interest and it provides an attractive strategy for designing artificial light-harvesting system. The manuscript could be accepted for publication after the following points are addressed:

1. There are some experiments were conducted in solution but didn't give the specific solvents.

2. TCS was introduced as pendant groups to the polyacrylate backbone, how to determine the ratio of NiR and TCS?

3. The authors mentioned “electronic ET” frequently in the manuscript. However, “electronic ET” is not well-known in the research of artificial light-harvesting system. So how to define the concept of “electronic ET”?

The electronic ET is considered as the reason for high efficiency. How to exclude the possibility of FRET? Is that explanation reasonable?

In page 10, the electronic ET was only evidenced by fluorescence anisotropy. The evidence is insufficient and not plausible. In addition, when the PTCS was dispersed in PMMA, columnar assembly may be disrupted, which should decrease the anisotropy. Why the anisotropy was high here? Are there any mistakes about the explanations? The same question for the well-ordered columnar assembly.

4. In page 9, line 153, “The marginal alterations observed in the spectra can be attributed to the feeble electronic interactions arising from the co-stacking of NiR and TCS in columns.” The red shift of emission wavelength may be caused by spectral overlap of donors and acceptors. When the doping amount of NiR is low, the spectrum of donor influences the spectral shape of acceptor a lot. Upon the doping amount of NiR increases, the emission intensity of donor decreased, resulting in the red shift described in the main text.

5. In page 12, LC and crystalline states of MTCS-NiR were discussed. What is the preparation process of the crystalline states of MTCS-NiR? How to ensure NiR molecules have been grown in the crystal rather than expelled by MTCS?

6. In Supplementary Figure 25, there is a mistake in legend which may confuse readers.

7. In page 12, line 242, “the intercolumnar correlation of the supramolecular columns induced by the incorporation of polymer main chain also contributes to the light-harvesting efficiency”. The author compared PTCS and MTCS in the LC state to explain this viewpoint. However, the arrangements of TCS units in PTCS columns and MTCS columns are different. How to prove the enhanced efficiency was caused by intercolumnar correlation but not the different arrangements of TCS units?

8. In Figure 4b, the illustration shows that intracolumnar crosslinking occurs between two PTCS polymer chains. However, the description in the main text means that it occurs in the same chain. That discrepancy may cause confusion.

9. The types of LHS reviewed in introduction were not comprehensive enough. Except the structural scaffolds mentioned, LHS based on supramolecular polymers (e.g., *Chem. Mater.*, 2019, 31, 3573–358) and nanocrystalline (e.g., *Angew. Chem. Int. Ed.* 2016, 55, 2759–2763) have been reported, which could be cited to complement the introduction.

10. The format of references is inconsistent. Please check and unify the format.

Response to Reviewers

Reviewer #1 (Remarks to the Author):

This work by Mu et al. introduces a new range of functional supramolecular materials exploiting a FRET process based on LC structures where a dye can intercalate to achieve red, blue or green colours, depending on the interaction of the dye with the LC assemblies.

The synthesis of the materials is well described and seems solid, with good evidence that the molecules of interest are indeed obtained. There is no doubt the luminescent study is very thorough and very convincing, showing the excellent luminescent properties of the systems.

Response: We appreciate for the kind remarks and encouraging comments.

1. I feel the authors should justify their choice of system better - the parallel with purple photosynthetic bacterial is to me very obscure, and the link seems to me very weak. Surely the authors can find examples of other, natural or synthetic, FRET systems based on supramolecular assemblies which exploit light harvesting to emit a range of colours. For instance, see:

Ahn, T. K.; Avenson, T. J.; Ballottari, M.; Cheng, Y.-C.; Ni-yogi, K. K.; Bassi, R.; Fleming, G. R., Architecture of a charge-transfer state regulating light harvesting in a plant antenna protein. *Science* 2008, 320 (5877), 794.

Wei, X.; Su, X.; Cao, P.; Liu, X.; Chang, W.; Li, M.; Zhang, X.; Liu, Z., Structure of spinach photosystem II–LHCII super-complex at 3.2 Å resolution. *Nature* 2016, 534 (7605), 69-74.

Son, M.; Pinnola, A.; Bassi, R.; Schlau-Cohen, G. S., The electronic structure of lutein 2 is optimized for light harvesting in plants. *Chem* 2019, 5 (3), 575-584.

Zhang, D.; Liu, Y.; Fan, Y.; Yu, C.; Zheng, Y.; Jin, H.; Fu, L.; Zhou, Y.; Yan, D., Hierarchical self-assembly of a dandelion-like supramolecular polymer into nanotubes for use as highly efficient aqueous light-harvesting systems. *Adv. Func. Mater.* 2016, 26 (42), 7652-7661.

Sarkar, A.; Behera, T.; Sasmal, R.; Capelli, R.; Empereur-mot, C.; Mahato, J.; Agasti, S. S.; Pavan, G. M.; Chowdhury, A.; George, S. J., Cooperative supramolecular block copolymerization for the synthesis of functional axial organic heterostructures. *J. Am. Chem. Soc.* 2020, 142 (26), 11528-11539.

Song, Q.; Goia, S.; Yang, J.; Hall, J.C.L.; Staniforth, M.; Stavros, V.G.; Perrier, S. Efficient

Artificial Light-Harvesting System Based on Supramolecular Peptide Nanotubes in Water, *J. Am. Chem. Sci.*, 2021, 143 (1), 382-389.

Kim, H.-J.; Nandajan, P. C.; Gierschner, J.; Park, S. Y., Light-harvesting fluorescent supramolecular block copolymers based on cyanostilbene derivatives and cucurbit[8]urils in aqueous solution. *Adv. Func. Mater.* 2018, 28 (4), 1705141.

Response: We appreciate your thoughtful evaluation of the manuscript. We recognize the differences in functions between these supramolecular column-based light-harvesting system and purple photosynthetic bacteria. In the natural process occurring in purple photosynthetic bacteria and higher plants, light-harvesting antennas absorb the solar energy and channel it through long-range energy funnels to reaction centers. These reaction centers facilitate conversion of transferred light energy into chemical energy. However, photosynthesis, from capturing solar energy to forming higher carbon products, is a complex, multistep process challenging to replicate artificially. Thus, mimicking certain photosynthetic steps aids our understanding and enhances solar energy utilization efficiency. Among these steps, light-harvesting stands out as pivotal, determining solar energy utilization efficiency. Therefore, substantial efforts have been directed toward developing artificial light-harvesting systems (such as Li, Q. et al. *Nanoscale Adv.* 2023, 5, 1830-1852; Yang, H.-B. et al. *Adv. Optical Mater.* 2020, 8, 2000265; Patra, A. et al. *Chem. Rev.* 2017, 117, 712-757; Yang, Q.-Z. et al. *Chem. Rev.* 2015, 115, 7502-7542). Addressing the reviewer's valuable input, the paper (*J. Am. Chem. Soc.* 2021, 143, 382-389) has already been cited as Ref. 38. Additionally, two closely related papers (*Adv. Funct. Mater.* 2016, 26, 7652-7661; *Adv. Funct. Mater.* 2018, 28, 1705141) on artificial light-harvesting systems have also been cited as Ref. 20 and 21 in this revised manuscript.

Our manuscript's aim is to mimic the ordered columnar packing assembly and its efficient excitation energy transfer mechanism in purple photosynthetic bacteria. Despite structural variances between our supramolecular columns and the ring-shaped columnar aggregates in purple photosynthetic bacteria, experimental results confirm the effectiveness of ordered columnar assemblies in constructing artificial light-harvesting systems. Thus, developing supramolecular column-based light-harvesting system inspired by the purple photosynthetic bacteria seems reasonable. In fact, many artificial light-harvesting systems draw inspiration

from natural systems and are applied to create diverse multi-color emissive materials (such as Yang, Q.-Z. et al. *Angew. Chem. Int. Ed.* 2016, 55, 2759-2763; Wang, F. et al. *Nat. Commun.* 2022, 13, 3546; Perrier, S. et al. *J. Am. Chem. Soc.* 2021, 143, 382-389; Stang, P. J. et al. *J. Am. Chem. Soc.* 2019, 141, 14565-14569).

To enhance reader understanding, we have clarified the comparative analysis in the revised manuscript: “Drawing inspiration from this natural process, we seek to organize chromophores into densely packed arrays to mimic the excitation ET process. In this regard, supramolecular columns with efficient intracolumnar electronic interactions may be ideal candidates, although the molecular arrangement between the chromophores and resulting columnar morphology differs from that in natural systems.” (pages 3-4)

2. I am not too clear why comparing a single TCS structural molecule with a polymer-conjugated TCS, and why choosing a fluorinated acrylate? Stability of the structures? Solubility? Polarity?

Response: We appreciate your insightful comments on the manuscript. The comparison of PTCS with MTCS was conducted based on two aspects. Firstly, MTCS is a symmetrical discotic molecule with a clearly identifiable columnar assembly structure, a feature that significantly aids in recognizing the columnar superlattice of PTCS, as detailed in the main text: “PTCS and MTCS exhibit two distinct types of Col_h phases, characterized by different hexagonal lattice parameters, despite sharing the same discotic mesogens (Fig. 3f).....In contrast, the lattice parameter of PTCS ($a = 4.63$ nm) was found to be approximately $\sqrt{3}$ times larger than that of MTCS, and the (11) peak was much stronger than (10) peak (Fig. 3f). This indicated the formation of a hexagonal columnar superlattice.....The columnar bundle was generated by the aggregation of main chains and further distribution on the basis of a hexagonal lattice (Fig. 3g). This packing arrangement reduced the symmetry to trigonal, resulting in a $p3m1$ plane group⁶³⁻⁶⁴.” (pages 13-14). Secondly, when assessing the light-harvesting performance of PTCS and MTCS, we deduced the polymeric effect on improving energy transfer efficiency, as discussed in the main text: “Moreover, the incorporation of polymer main chain induces intercolumnar correlation within the supramolecular columns, a significant factor contributing to the light-harvesting efficiency.

This can be demonstrated by a comparative study of PTCS and MTCS in the LC state.....”
(page 13).

The incorporation of TCS into a polymer system to create PTCS aims to improve inter-chromophore correlations within the columnar phases, thereby promoting excitation energy migration. This strategy stems from our extensive experience with discotic liquid crystalline polymers (*Macromolecules* 2015, 48, 6768-6780; 2015, 48, 2388-2398; 2019, 52, 6913-6926). We have previously shown that incorporating discotic liquid crystals into a suitable polymer system not only facilitates the formation of an ordered columnar superlattice structure but also enhances intercolumnar correlations. To achieve this, we selected a polyacrylate backbone based on our familiarity with discotic liquid crystalline polymers, appreciating its flexibility. The polyacrylate-based discotic liquid crystal polymer effectively balances the columnar stacking ability of pendant discotic TCS mesogens and accommodates the polymer main chain within intercolumnar channels, leading to the formation of columnar superlattices. However, despite our attempts to prepare PTCS directly through free radical polymerization of the TCS-based acrylate monomer, no polymer was obtained, even with varying solvents, temperatures, and concentrations. Thus, we adopted an indirect two-step synthesis pathway, following a methodology reported in the literatures (Das, A. & Theato, P. *Macromolecules* 2015, 48, 8695-8707; *Chem. Rev.* 2016, 116, 1434-1495). In this approach, we first synthesized poly(pentafluorophenyl acrylate) through free radical polymerization of commercially available pentafluorophenyl acrylate. Subsequently, TCS units were efficiently attached to the polyacrylate backbone via nearly quantitative trans-esterification of poly(pentafluorophenyl acrylate) and mono-hydroxyl-functionalized TCS. As expected, the resulting PTCS formed an ordered columnar superlattice assembly with enhanced intercolumnar correlations and demonstrated highly efficient light-harvesting performance.

3. The characterisation of the assembled structures could be more through - although the authors demonstrate the LC properties, I am not too clear why they expect a stacked / columnar structure. Do they obtain individual rod-like nanoparticles in solution? Could the authors image these structures using TEM or AFM, or characterise them using scattering techniques such as SANS?

Response: We appreciate your insightful comments on the manuscript. The reviewer's suggested additional characterizations in solution employing TEM, AFM, or SANS can indeed provide more direct evidence in real space, such as individual rod-like nanoparticles. However, these techniques appear not suitable for our system, because there are no obvious aggregation behaviors observed in the TCS based monomers and polymers in solution. Alternately, we identify the columnar liquid crystal structures through characteristic POM textures and synchrotron radiation small-angle X-ray scattering (SAXS) analysis. These techniques are commonly used for liquid crystal phase identification. POM revealed a pseudo focal conic fan-shaped texture for MTCS and an atypical texture for PTCS due to the small domains of LC polymers (Supplementary Fig. 9). The SAXS patterns displayed the $1:\sqrt{3}$ ratio of reciprocal spacings, typical of two-dimensional hexagonal columnar lattices (Supplementary Fig. 10). Furthermore, the presence of columnar structures is reinforced by our previous publications concerning TCS-based columnar liquid crystals, particularly the single crystal columnar structures of TCS analogs (Ref. 50: *ACS Appl. Mater. Interfaces* 2020, 12, 9637-9645; Ref. 51: *Chem. Eur. J.* 2023, 29, e202300320), which provide unambiguous evidence. These descriptions have been incorporated into the revised manuscript: "*Both PTCS and MTCS formed thermotropic hexagonal columnar (Col_h) LC phases due to the discotic shape of the TCS core with aliphatic surroundings, which was reminiscent of the columnar structures generated by TCS derived LC compounds^{50,51}.*" (page 6) In our modest opinion, the presented experimental data, as well as the support from our previous works, are sufficient to identify the columnar structures.

4. Overall I think the work is novel and could be published in Nature Comm, the luminescent properties are remarkable, but the authors should better justify their choice of materials, and better characterise the supramolecular structure they obtain.

Response: We appreciate your thorough evaluation of the manuscript. Your positive assessment of the novelty and significance of the work is encouraging. In response to the reviewer's feedback, we have made specific revisions, such as clarifying the choice of the materials (in response to comment 1) and providing additional support for the supramolecular columnar structure (in response to comment 3). These revisions were made to effectively

address the comments and concerns raised by the reviewer.

Reviewer #2 (Remarks to the Author):

The authors present their studies in discotic molecular assemblies in facilitating long distance energy transfer inspired by light harvesting antenna in natural photosynthesis. They claimed that observation of large antenna effect in the assemblies and carried out variation of the acceptor/chromophore ratios as well as environmental modifications on the rigidity of the assemblies. They used time resolve fluorescence lifetime and anisotropy measurements to support their claims. Finally, they showed the chemically tunability of the emissions and proposed applications of these properties.

I recommend its publication of revised manuscript in a materials focused journal rather than Nature Communications for the reasons below:

Response: We appreciate your thorough evaluation of the manuscript. This work focuses on developing a highly efficient artificial light-harvesting system using discotic columnar liquid crystalline polymers. The designed discotic liquid crystalline polymer exhibits high-efficiency fluorescence, meeting the requirements for constructing artificial light-harvesting systems. Its ordered columnar structure facilitates highly efficient excitation energy transfer to the acceptor. The experimental results confirm the effectiveness of this modularly assembled columnar system, demonstrating an efficient yet controllable energy transfer process. This approach opens new avenues for the development of efficient artificial photosynthesis systems.

Specially, the manuscript delves into the design and synthesis of fluorescent discotic liquid crystalline polymer, the study of liquid crystal structures through supramolecular self-assembly, the photophysical properties of the liquid crystalline polymer, excitation energy transfer mechanisms for light-harvesting, and their applications in information encryption. This multidisciplinary work spans polymer chemistry, supramolecular chemistry, photochemistry, photophysics, and material science. Notably, similar publications focusing on artificial light-harvesting systems and their fluorescent functionalities have appeared in the journal (such as Wang, F. et al. *Nat. Commun.* 2022, 13, 3546; Tao, S. et al. *Nat. Commun.* 2022, 13, 6034; Li, Q. et al. *Nat. Commun.* 2023, 14, 3005; Andreasson, J. et al. *Nat. Commun.*

2019, 10, 3996). Considering the above aspects, we believe this manuscript is well-suited for consideration to be published in the general journal of *Nature Communications*.

1. The manuscript is not written with a high standard of scholarship with poor flow and repeated statements. Some of the concepts, such as antenna effect, were not defined. Some of the claims are based on hotchpotch results without firm and convincing analyses.

Response: We appreciate your detailed review of the manuscript. In response to your feedback, we have made substantial efforts to enhance the clarity and coherence of the text, as highlighted in the revised manuscript. Additionally, a native English-speaking expert has further polished the language throughout the manuscript. Recognizing the importance of defining key concepts, we have now included a sentence in the introduction to explain the antenna effect: “*Notably, the ET was detectable even at a high donor/acceptor ratio of 20000:1, and the antenna effect (AE, a factor that describes how much brighter the acceptor emits when excited by the donor instead of being directly excited) was ultrahigh, exceeding 100.*” (page 4). To provide more firm and convincing analyses, we revisited our data and conducted additional analyses to support our claims, as highlighted in the revised manuscript. These revisions significantly enhance the manuscript’s quality, addressing your concerns regarding writing standards, concept definition, and data analysis.

2. The application of discotic liquid crystal materials for light harvesting has been in the literature for decades. Thus, the concept is not novel. Peoples have used porphyrins, phthalocyanines, cronenes and other extended planar aromatic groups to form the columnar structure to conduct charges and facilitate energy transfer.

Response: We appreciate your careful assessment of the manuscript. Regarding your concerns about the work’s novelty, it is essential to emphasize that this submission explores fluorescent discotic liquid crystals and their implications for light-harvesting applications, distinguishing them from conventional semiconducting discotic liquid crystals where fluorescence is typically quenched. Discotic liquid crystals, based on porphyrins, phthalocyanines, cronenes and other extended planar aromatic groups, have been extensively studied. They easily stack into columnar structures with intracolumnar π - π interactions,

facilitating 1D charge carrier migration pathways along the columnar axis, making them valuable for organic electronics (such as Geerts, Y. H. et al. *Chem. Soc. Rev.* 2007, 36, 1902-1929; Li, Q. et al. *Prog. Mater. Sci.* 2019, 104, 1-52). In this context, we utilized triphenylene and perylene bisimide units as the discotic mesogens to prepare various discotic liquid crystalline polymers and further investigated the charge mobility in their columnar phases (*Macromolecules* **2023**, 56, 4845-4854; **2019**, 52, 6913-6926; **2015**, 48, 6768-6780; **2015**, 48, 2388-2398). However, the molecular fluorescence inherent to the π -conjugated nature is strongly quenched in this case, hindering their applications in constructing artificial light-harvesting systems. On the other hand, although these discotic units have been used to construct various supramolecular scaffolds for light-harvesting, the implication of energy transfer is usually studied in molecularly isolated state, limiting the light-harvesting efficiency due to the absence of efficient intermolecular interactions necessary for promoting excitation energy transfer (such as Zhou, Y. et al. *Angew. Chem. Int. Ed.* 2016, 55, 7952-7957; Perrier, S. et al. *J. Am. Chem. Soc.* 2021, 143, 382-389; Zhang, M. et al. *J. Am. Chem. Soc.* 2020, 142, 18763-18768). Based on these considerations, we designed a TCS-based discotic liquid crystalline polymer, which stacks into columnar arrangements while maintaining bright fluorescence in their columnar phases. The highly efficient solid-state fluorescence meets the requirements for artificial light-harvesting, and the ordered columnar arrangement provides excitation energy migration pathways. This unique combination results in a highly efficient artificial light-harvesting system. To the best of our knowledge, this is the first example of artificial light-harvesting system based on discotic liquid crystalline polymers, thus supporting the novelty and significance of the work for publication in *Nature Communications*.

3. The authors need to have an excitation fluence dependent study to determine the “antenna effects” and energy donor/acceptor ratio to make sure that the effect is not due to multiple exciton generation in the system. It is important to quantify the quantum yield of energy transfer.

Response: Thank you for your valuable feedback. Based on your suggestion, we conducted the excitation power dependent fluorescence study (Supplementary Fig. 16). The linear

dependence with the slope around 1 provides evidence for the single photon energy transfer dynamics and confirms the absence of multiple exciton generation. These results are now included in the revised manuscript: “*Additionally, fluorescence spectra recorded under varying excitation power revealed a linear intensity dependence for both donor and acceptor emissions with a slope of about 1 throughout the entire excitation range (Supplementary Fig. 16). This linear dependence proves that ET in the supramolecular column-based LHS occurs due to a single photon event, eliminating the possibility of multiple exciton processes.*” (page 7). Based on the understanding of this single photon event, we were able to determine the quantum yield of energy transfer, as discussed in the revised manuscript: “*This process also correlated with an increase in quantum yield of ET (or ET efficiency), reaching nearly 100%,...*” (page 9).

4. The authors were inspired by natural photosynthetic light harvesting systems with chlorophyll arrays, however they ignored an important aspect of this array where the chlorophylls are arranged with a propeller fashion instead of well-aligned face-to-face conformation as in the columnar assemblies in their study. Many self-assembled systems are very sensitive to the alignment of the atoms between the chromophores. It has been well documented that for energy or charge transfer, some off-set between the pi-stacked entities of the same species could have superior properties in energy and charge transfer and transport. Although the authors indicated that they tried to avoid the quenching of the exciton by adapting a twisted conformation, but inter-chromophore interactions were not carefully characterized.

Response: We appreciate your insightful comments on the manuscript. We acknowledge the distinction between the propeller arrangement of chlorophylls in natural photosynthetic systems and the helically stacked columnar assemblies in our study, which may exert certain influence on their energy transfer process. However, replicating the intricate assemblies observed in natural systems proves challenging. Despite structural differences from purple photosynthetic bacteria, our manuscript aims to mimic the ordered molecular stacking assembly mode and its efficient excitation energy transfer mechanism. The results of our work demonstrate that these ordered columnar structures indeed substantially enhance

light-harvesting efficiency. In fact, many bioinspired light-harvesting systems have referred to the ordered molecular arrangements in natural system rather than precisely replicating them (Such as Yang, Q.-Z. et al. *Angew. Chem. Int. Ed.* 2016, 55, 2759-2763; Wang, F. et al. *Nat. Commun.* 2022, 13, 3546; Sargent, E. H. et al. *Nat. Rev. Mater.* 2020, 5, 828-846).

We recognize the importance of characterizing inter-chromophore interactions more comprehensively in our supramolecular columnar systems. The twisted three-armed molecular geometry adopting a helical stacking fashion in columns hinders the precise characterization of the inter-chromophore distance via X-ray scattering (such as Kato, T. et al. *J. Am. Chem. Soc.* 2012, 134, 5652-5661; Sierra, T. et al. *J. Am. Chem. Soc.* 2006, 128, 4487-4492). Alternately, we conducted additional experimental analyses of their photophysical properties: “Both absorption and emission spectra of PTCS experienced a bathochromic shift in the solid state compared to the solution state (Supplementary Fig. 13), indicating J-type aggregation of the discotic TCS units within the columns.” (page 7). Moreover, NiR/TCS ratio dependent fluorescence studies (Figs. 2e,f), fluorescence anisotropy studies (Figs. 3a,b), and also theoretical calculations (Fig. 3c) provide strong evidence for inter-chromophore interactions within the columns. To evaluate the exciton dynamics in the columnar structures, we additionally determined the exciton migration rate constant as presented in Supplementary Fig. 21. The corresponding discussions are included in the revised manuscript: “Moreover, the exciton migration rate constant was estimated to be $1.65 \times 10^{11} \text{ L mol}^{-1} \text{ s}^{-1}$ (Supplementary Fig. 21), indicating significantly faster migration of exciton energy within the columns compared to the diffusion limit for bimolecular reactions in solution^{31,56}.” (page 10). All these evidences support the inter-chromophore interactions within the ordered columnar liquid crystal assembly, significantly enhancing the excitation energy transfer efficiency.

5. The authors claimed that “the electronic coupling of stacked aromatic chromophores in resulting supramolecular columns generates coherent energy diffusion” but the concept of coherent energy transfer was not elaborated and proven for the systems in the study.

Response: Thank you for your professional feedback. Initially, we employed the concept of coherent electronic energy transfer, drawing from the inter-chromophore electronic

interactions and fluorescence anisotropy decrease within the columnar structures. Coherent energy transfer has been largely studied in chromophore ensembles isolated from natural light harvesting complexes, as well as in conjugated polymer or molecular aggregates of highly defined sizes and geometries (Scholes, G. D. et al. *Science*, 2009, 323, 369-373; Kim, D., Würthner, F. et al. *J. Am. Chem. Soc.* 2018, 140, 4253-4258). However, acquiring direct evidence within our dynamic liquid crystal system, which features undefined extended aggregates where size and exact packing are typically not well characterized, has proven challenging. Taking into account your feedback and comment 3 from reviewer #3, we reevaluated the energy transfer process. It became evident that the current experimental data do not offer sufficient support for the concept of coherent energy transfer. At least, we cannot rule out the possibility of Förster resonance energy transfer. Therefore, in the revised manuscript, we have rephrased the explanation, opting for the general term “excitation energy transfer” instead of “coherent energy transfer”. The experimental results in our manuscript, including light-harvesting study (Fig. 2) and fluorescence anisotropy analysis (Fig. 3b), robustly support the concept of excitation energy transfer. Notably, excitation energy transfer has been widely utilized in various artificial light-harvesting systems (such as Tung, C.-H., Yang, Q.-Z. et al. *Chem. Rev.* 2015, 115, 7502-7542; Scholes, G. D. et al. *Chem. Rev.* 2017, 117, 249-293).

6. The emission anisotropy measurements have serious flaws. In figure 3b, the authors compared the anisotropy of the isolated chromophore in a rigid PMMA matrix which has a much higher anisotropy due to a fixed orientation of the transition dipole moment for the emission to the chromophore with Nile red in columnar structure. It will be useful to just measure the columnar chromophore only without Nile red as a reference. According to the authors that the columnar structure has a spiral orientation between pi-stacked molecules. Thus, even without Nile red, it could have energy diffusion from one chromophore to another and have lower anisotropy compared to the isolated chromophore in the solid matrix. If the insertion of Nile red speeds up the energy migration following the Förster mechanism, the decay of anisotropy would be faster. Without the comparison with the neat columnar system, the study is in question. Meanwhile, I am not convinced on the small change in the system

with Nile Red. What are the error bars in those measurements?

Response: We appreciate your detailed review of the manuscript. In response to your comments, we have made significant improvements to our experimental approach. Specifically, we included anisotropy measurement for the neat columnar chromophore of PTCS. Moreover, we repeated the fluorescence anisotropy experiments and included error analysis for the anisotropy measurements. All relevant data have been summarized in Fig. 3b. As expected by the reviewer, the results indicate a significant reduction in the anisotropy of the columnar PTCS compared to that dispersed in PMMA, demonstrating efficient excitation energy migration via the columnar structures. The intercalation of NiR into these supramolecular columns further decreased anisotropy, as it accelerated the migration of excitation energy within the system. However, it is noteworthy that weak alterations were observed when higher NiR/TCS ratios were used. This minor reduction is probably due to the fact that the determined anisotropy of 0.006 is already quite low (near zero), making it challenging to precisely define the further decrease in anisotropy due to instrumental limitations. Notably, the nearly zero fluorescence anisotropy exhibited a larger standard error than the observed anisotropy value, due to the rapid redistribution of the excitation energy over the columns (similar to observations in literature such as Klymchenko, A. S. et al. *Adv. Mater.* 2023, 35, 2301402). The corresponding experimental results and analysis have been revised in the manuscript: *“Indeed, experimental results demonstrated a 15-fold reduction in fluorescence anisotropy from 0.24 to 0.016 within the columnar assembly (Fig. 3b and Supplementary Figs. 19-20). The intercalation of NiR into these supramolecular columns accelerated the migration of excitation energy within the system, leading to a remarkable decrease in anisotropy to an impressive 0.006. However, it is worth noting that achieving such a low anisotropy level poses a challenge in precisely defining its further decrease with increasing NiR/TCS ratios due to instrumental limitations. The nearly zero fluorescence anisotropy exhibited a larger standard error than the observed value, suggesting that the excitation energy redistributed rapidly within the columns.”* (page 10).

Reviewer #3 (Remarks to the Author):

The authors present the construction of a supramolecular assemblies for mimicking natural

light-harvesting system of purple photosynthetic bacterial. A discotic columnar liquid crystalline polymer was selected as an energy donor and NiR served as an energy acceptor. NiR intercalated into the polymeric columnar liquid crystal to form the supramolecular artificial light-harvesting system. The highly ordered packing of chromophores in such light-harvesting system based on liquid crystalline polymer facilitates the efficient energy transfer from energy donors to acceptors. It is impressive that the energy transfer can be observed even at ultrahigh D/A ratio of 20000:1. The emission colors of the light-harvesting systems can be tuned by changing the ratio of D to A. The light-harvesting systems with different emission color have been applied for information encryption with spatiotemporal regulation security. Overall, the results presented in this paper is of great interest and it provides an attractive strategy for designing artificial light-harvesting system. The manuscript could be accepted for publication after the following points are addressed:

Response: We appreciate for the kind remarks and encouraging comments.

1. There are some experiments were conducted in solution but didn't give the specific solvents.

Response: We appreciate your thorough review of the manuscript. In response to your feedback, we have included comprehensive information about the solvents used in the experiments. This information is now available in Supplementary Figs. 13, 14, 26, 27, 29.

2. TCS was introduced as pendant groups to the polyacrylate backbone, how to determine the ratio of NiR and TCS?

Response: We appreciate your insightful comment on the manuscript. The ratio of NiR to TCS can be directly determined by comparing NiR to the repeat unit of PTCS, a value ascertainable through their feeding ratios. The almost quantitative trans-esterification ensured the complete attachment of the TCS units to each repeat unit of the polyacrylate backbone, facilitating the direct determination of the amount of TCS units via the ratio of the PTCS's weight to the molecular weight of the corresponding repeat unit. Similarly, the amount of NiR is directly determined by its weight and molecular weight. Both factors lead to the simple determination of the NiR/TCS ratio through their predesigned weights and known molecular

weights. Accordingly, the corresponding description has been included in the revised manuscript: “*This method ensured almost quantitative efficiency (Supplementary Figs. 4-7), facilitating precise determination of the amount of TCS units incorporated into the PTCS.*” (page 6).

3. The authors mentioned “electronic ET” frequently in the manuscript. However, “electronic ET” is not well-known in the research of artificial light-harvesting system. So how to define the concept of “electronic ET”? The electronic ET is considered as the reason for high efficiency. How to exclude the possibility of FRET? Is that explanation reasonable? In page 10, the electronic ET was only evidenced by fluorescence anisotropy. The evidence is insufficient and not plausible. In addition, when the PTCS was dispersed in PMMA, columnar assembly may be disrupted, which should decrease the anisotropy. Why the anisotropy was high here? Are there any mistakes about the explanations? The same question for the well-ordered columnar assembly.

Response: We appreciate your thoughtful review of the manuscript. Initially, we employed the concept of coherent electronic energy transfer, drawing from the inter-chromophore electronic interactions and the decrease in fluorescence anisotropy within the columnar structures. However, after considering your feedback and comment 5 from reviewer #2, we acknowledged the lack of substantial evidence to specifically define electronic energy transfer, excluding the possibility of Förster resonance energy transfer at the present stage. Therefore, in the revised manuscript, we have replaced this concept with the more widely accepted term “excitation energy transfer”, aligning with relevant references on artificial light-harvesting systems (such as Tung, C.-H., Yang, Q.-Z. et al. *Chem. Rev.* 2015, 115, 7502-7542; Scholes, G. D. et al. *Chem. Rev.* 2017, 117, 249-293). Our experimental findings, including the light-harvesting study (Fig. 2) and fluorescence anisotropy analysis (Fig. 3b), strongly support the concept of excitation energy transfer.

Regarding the high anisotropy of the isolated chromophore in a rigid PMMA matrix, it resulted from a fixed orientation of the transition dipole moment for the emission to the chromophore without the formation of columnar structure (Genovese, D. et al. *Nanoscale* 2014, 6, 3022-3036). In contrast, a low anisotropy was observed in neat PTCS due to energy

diffusion from one chromophore to another within the stacked columnar structures (Fig. 3a,b). Thus, it is reasonable to infer that neat PTCS exhibits lower anisotropy compared to the isolated chromophore in the solid PMMA matrix.

4. In page 9, line 153, “The marginal alterations observed in the spectra can be attributed to the feeble electronic interactions arising from the co-stacking of NiR and TCS in columns.” The red shift of emission wavelength may be caused by spectral overlap of donors and acceptors. When the doping amount of NiR is low, the spectrum of donor influences the spectral shape of acceptor a lot. Upon the doping amount of NiR increases, the emission intensity of donor decreased, resulting in the red shift described in the main text.

Response: We appreciate your careful review of the manuscript. In the revised version, we have clarified the emission peak of NiR by eliminating the contribution from TCS, as demonstrated in Supplementary Fig. 18. Additionally, even when excited at the maximum absorption of NiR around 530 nm, the emission exhibited a similar trend with the variation of NiR/TCS (Supplementary Fig. 15). Thus, we believe it is reasonable to attribute the observed spectral alterations to the feeble electronic interactions arising from the co-stacking of NiR and TCS in columns.

5. In page 12, LC and crystalline states of MTCS-NiR were discussed. What is the preparation process of the crystalline states of MTCS-NiR? How to ensure NiR molecules have been grown in the crystal rather than expelled by MTCS?

Response: We appreciate your insightful comment on the manuscript. The preparation process involved mixing a specific quantity of NiR and MTCS (or PTCS) in a solution, followed by slow solvent evaporation, yielding the corresponding solid samples in either crystalline or LC states. To investigate the intercalation of NiR molecules into the crystal lattice or expelled by MTCS, we additionally conducted X-ray scattering experiments. These results have been presented in Supplementary Fig. 25 to support the assertion that NiR molecules are indeed a part of the crystalline or LC state: “*After intercalation of NiR, both the crystalline columnar and Col_h LC phases can be largely maintained, suggesting the co-stacking of NiR and MTCS into homogeneous columnar structures.*” (page S21).

6. In Supplementary Figure 25, there is a mistake in legend which may confuse readers.

Response: Thank you for your careful review of the manuscript. We have now corrected the legend to accurately describe the content of the figure, eliminating any potential confusion for readers.

7. In page 12, line 242, “the intercolumnar correlation of the supramolecular columns induced by the incorporation of polymer main chain also contributes to the light-harvesting efficiency”. The author compared PTCS and MTCS in the LC state to explain this viewpoint. However, the arrangements of TCS units in PTCS columns and MTCS columns are different. How to prove the enhanced efficiency was caused by intercolumnar correlation but not the different arrangements of TCS units?

Response: We appreciate your thoughtful review of the manuscript. The reviewer made an insightful observation regarding the potential impact of different TCS unit arrangements within PTCS and MTCS columns on their efficiency. Our structural characterization results confirm that both PTCS and MTCS self-assemble into columnar liquid crystal phases through the stacking of TCS units. While PTCS and MTCS exhibit columnar phases with different plane groups, $p3m1$ and $p6mm$, respectively, it is noteworthy that their intracolumnar stacking arrangements are remarkably similar. This similarity is supported by their wide-angle X-ray scattering profiles (Supplementary Fig. 10). The key difference between them lies in the accommodation of PTCS’s main chains within the intercolumnar channels, resulting in a columnar superlattice that enhances the intercolumnar correlations. Moreover, as liquid crystals are dynamic molecular assemblies, the TCS units within the columns exhibit a certain degree of motion (Müllen, K. et al. *Adv. Mater.* 2010, 22, 3634-3649), further minimizing the arrangement differences between PTCS and MTCS. Given the similar intracolumnar arrangements, we believe it is reasonable to deduce that the light-harvesting efficiency difference primarily originates from additional intercolumnar correlations.

8. In Figure 4b, the illustration shows that intracolumnar crosslinking occurs between two PTCS polymer chains. However, the description in the main text means that it occurs in the same chain. That discrepancy may cause confusion.

Response: We appreciate your careful review of the manuscript. We agree with the reviewer's observation that intracolumnar crosslinking occurs between different PTCS polymer chains, as illustrated in Fig. 4b. The corresponding description in the main text "*It is noteworthy that due to the columnarly stacked TCS surrounded by alkyl shells, light-induced cycloaddition was confined within the intracolumnar cores (Supplementary Fig. 34). Intracolumnar crosslinking led to the expulsion of NiR from the TCS columns, substantially reducing the ET process due to the absence of co-stacked structure (Fig. 4b).*" (pages 14-15) clarifies that intracolumnar crosslinking occurs within the same column rather than the same polymer chain. Therefore, we believe it is reasonable and there is no discrepancy between the description and the content of Fig. 4b.

9. The types of LHS reviewed in introduction were not comprehensive enough. Except the structural scaffolds mentioned, LHS based on supramolecular polymers (e.g., *Chem. Mater.*, 2019, 31, 3573–358) and nanocrystalline (e.g., *Angew. Chem. Int. Ed.* 2016, 55, 2759–2763) have been reported, which could be cited to complement the introduction.

Response: We appreciate your valuable input regarding the comprehensiveness of the types of light-harvesting systems mentioned in the manuscript. In response to your suggestion, we have expanded the introduction section. We included the paper (*Chem. Mater.* 2019, 31, 3573-3581) as Ref. 30 in the nanoparticles, and the paper (*Angew. Chem. Int. Ed.* 2016, 55, 2759-2763) as Ref. 31 in an additional type of nanocrystals. This classification is based on structural scaffolds. These references will complement the existing introduction and provide readers with a broader perspective on the subject matter.

10. The format of references is inconsistent. Please check and unify the format.

Response: We appreciate your attention to detail regarding the formatting of references in the manuscript. We have meticulously reviewed and standardized the format of all references to ensure uniformity according to standard Nature referencing style.

REVIEWERS' COMMENTS

Reviewer #1 (Remarks to the Author):

In this manuscript by Mu et al, the authors have thoroughly addressed all comments from the reviewers. In particular they explanation of using NiR as intercalated element in the structures help a lot clarifying this aspect of the work, and the antenna effect study is a good addition to support the conclusions. I also note that the English language as been vastly improved.

Overall the authors have addressed all comments appropriately, and I therefore recommend the manuscript is now accepted for publication.

Reviewer #3 (Remarks to the Author):

I read through the revised manuscript and responses to the comments of reviewers, the issues raised by three reviewers have been addressed in the revised manuscript. I have also gone through the comments made by the reviewer #2, and the comments made by reviewer #2, particularly the concerns related to the novelty about the light-harvesting systems based on discotic liquid crystals might be too critical. I think the authors have performed sufficient additional experiments and explanations to address the concerns of reviewer #2. I am therefore satisfied to accept its publication in Nat. Commun.